# An Exploration of Psychological Resilience among Undergraduate Nursing Students Undertaking an Adult Nursing Virtual Practicum during the Coronavirus Pandemic in Taiwan: A Qualitative Study

**DOI:** 10.3390/ijerph20021264

**Published:** 2023-01-10

**Authors:** Chun-Chih Lin, Fred Arne Thorberg, Ya-Ling Huang, Chin-Yen Han, Ching-Ching Su, Li-Chin Chen

**Affiliations:** 1Department of Nursing, Chang Gung University of Science and Technology, No.2., Sec. W., Jiapu Rd., Puzi City 61363, Taiwan; 2New Taipei Municipal TuCheng Hospital, Chang Gung Medical Foundation, No.6, Sec. 2, Jincheng Rd., Tucheng Dist., New Taipei City 236017, Taiwan; 3School of Psychology, Bone University, 14 University Drive, Robina, Gold Coast, QLD 4226, Australia; 4Faculty of Health (Nursing), Southern Cross University, Gold Coast Campus B7.47, Coolangatta, Gold Coast, QLD 4225, Australia

**Keywords:** nursing students, psychological resilience, virtual practicum, coronavirus, Taiwan

## Abstract

This qualitative study aimed to explore the psychological resilience of undergraduate nursing students partaking in a virtual practicum during the coronavirus pandemic (COVID-19) in Taiwan. The virtual practicum, a form of online learning, creates challenges compared to the traditional teaching–learning experience of an actual clinical placement. Exploring how students overcome learning difficulties and build resilience is necessary for a new learning environment or for future online learning. Constructivist grounded theory and the Standards for Reporting Qualitative Research checklist were followed. Purposive and theoretical sampling were used to recruit 18 student nurses for data saturation. Semi-structured, face-to-face interviews were conducted individually to collect data. Initial, focused, and theoretical coding and constant comparative data analysis were performed. Credibility, originality, resonance, and usefulness guided the assessment of the study’s quality. The core category of psychological resilience in the virtual practicum was constructed to reflect Taiwanese nursing students’ progress and experiences of learning during the virtual practicum. This core category consisted of three subcategories: (i) learning difficulties within one’s inner self; (ii) staying positive and confident; and (iii) knowing what is possible. The findings identified psychological resilience as an important factor for students to adjust to the adverse experiences of a rapidly changing learning environment, such as the virtual practicum. The substantive theory of psychological resilience provided a frame of reference for coping with possible future difficulties. Correspondingly, psychological resilience reflected individuals’ potential characteristics and may help students to enter and remain in the nursing profession.

## 1. Introduction

The coronavirus pandemic (COVID-19) has been an ongoing global health crisis [1]. The number of confirmed cases of coronavirus disease has increased dramatically over time, and the coronavirus has repeatedly mutated to threaten human health. Many countries have suffered severely from the pandemic, despite the implementation of policies involving frequent handwashing, using sanitizers, wearing facial masks, and maintaining social distancing. Taiwan was also confronted by the coronavirus threat and introduced a zero-tolerance strategy to control the spread of the virus, which worked well until the coronavirus entered the community in May 2021. This wave of the outbreak had a major impact across all areas of peoples’ daily lives, including students’ learning.

Furthermore, all levels of Taiwan’s education system transitioned from teaching and learning in person to emergency remote education. Nursing education in Taiwan was no exception, and clinical placements had to adhere to the policy of a suspension of classes without postponing learning [2]. Students had no previous experience with remote learning in the context of clinical placements, such as the virtual practicum (VP). To the best of our knowledge, the VP’s influence on students’ adult nursing clinical placement was limited from the students’ perspectives [3], as well as how students experienced the VP process, as it challenged the traditional nursing practicum. Therefore, this study explored undergraduate nursing students’ experiences of their psychological resilience or lack thereof during a VP throughout the coronavirus pandemic.

If the coronavirus pandemic is an indelible catastrophe in human history, humans will never be satisfied that they have done enough to prepare for disasters [4]. However, good preparation for any unpredictable event can assist in quick response and reduce adverse consequences. The pandemic completely subverted the transformation of an online learning environment from traditional classroom settings, including nursing students’ clinical placements. Although both educators and students dealt with the challenges of online learning, this was easier to implement for didactic lectures compared to laboratory-based learning [5]. When an actual clinical practicum was not feasible, the VP provided an opportunity to continue learning despite the pandemic [6,7]. Accordingly, Alqahtani et al. [8] suggested that the nursing curriculum may need revision to provide reliable, effective alternative teaching and learning options compared to more traditional methods.

Previous research suggests that laboratory-based online learning, such as the VP, caused students to have learning anxiety, a lack of motivation, a decreased quality of learning and difficulties with communication, cooperation, and interaction, as well as an inability to practice real-world skills [9]. Moreover, these learning difficulties led to negative attitudes towards the VP [10]. Although studies have explored online learning, such as video learning for skills practice [11], the research focused on online technological learning, and the findings were inconsistent with respect to learning efficacy [12]. Clearly, VP learning is not about the use of advanced technology or software, but about the design of teaching and learning strategies. Therefore, it is important to view the VP learning experience from a student’s perspective.

It is generally believed that traditional teaching methods, such as lectures, can effectively teach complex theories and integrate multidisciplinary knowledge [13]. However, limited information exists regarding the effectiveness of virtual learning, especially in VPs. Experiential learning, such as clinical placements, is considered essential in the nursing field. Although clinical placements could be replaced with high-quality simulations, no substitute has been found for the high-quality simulation experience described by Hayden et al. [14]. James et al. [15] emphasised the importance of nursing students’ experiences and the need to adapt to unfamiliar learning environments, as well as learning to care for actual patients during their clinical training. Student nurses confronted dilemma, the opportunity of clinical placement or the challenge of negative mood, uncertainty, nervousness, and fear from the working environment [16]. This dilemma led student nurses to feel hesitant and doubtful about making care judgements; however, they learned by doing and constructed their psychological literacy. Thus, there is a need to understand how students develop psychological resilience during VPs.

## 2. Materials and Methods

### 2.1. Design

Grounded theory is derived from a social constructionist perspective on the interaction between individuals and society. It is based on symbolic interactionism (SI), which focuses on interaction and the meaning of events to define an individual’s situation [17]. SI emphasizes the processes of defining, acting, and using language on people’s interactions with the environment. The theoretical framework of SI describes how individuals interpret objects and events in their lives and the process of how their interpretations lead to action in a specific situation [18,19]. It also provides a perspective on understanding how people in a particular situation define reality.

Traditional grounded theory aims to discover and interpret people’s actions and interactions with their environment as a psychosocial process to understand an individual’s overt and covert behaviour [19]. In the grounded theory paradigm, data collection and analysis proceed concurrently; the constant comparative method and theoretical sensitivity to recognise meaningful data are emphasised. Although there have been epistemological and methodological differences within the grounded theory movement [18], grounded theory places an emphasis on obtaining meaningful insights systematically to develop a substantive or conceptual theory which can be used to explain the phenomena under investigation [19].

Constructivist grounded theory (CGT) served as the theoretical basis for this study. CGT seeks to achieve a balance between the objectivity and subjectivity of the study. Although a certain degree of subjectivity is unavoidable, the theoretical concept of the constructivist approach provides an abstract understanding rather than an explanation and prediction [20]. Knowledge is created and constructed through the study process, and this allowed for an inductive exploration of the complex process of adapting a virtual practicum for nursing students during the coronavirus pandemic. CGT uses an inductive approach to generate an understanding of social phenomena. As part of our society, we get involved, interact, and share perspectives with the phenomena that we study. As such, this investigation was constructed from the participants’ reality, their implicit meanings, and their experiential views from the VP. The study adopted CGT as a set of practices that followed certain methods, starting with data gathering and ending with analysis writing and reflection on the entire process [20]. This work was presented according to the Standards for Reporting Qualitative Research (COREQ) checklist [21].

### 2.2. Research Group and Ethical Considerations

The study’s participants (as a social phenomenon) were selected from an undergraduate nursing programme at a university in southern Taiwan. Potential participants received an invitation through the university email system and, if interested, responded to the first author. A total of 150 students are enrolled in this programme annually. By May 2021, the coronavirus was widespread in the community and thus interfered with the second-year nursing students’ regular clinical placement in Adult Nursing. Those students who undertook the VP later returned to school in September 2021 as the new semester commenced. Recruiting criteria for research group were: (i) 20 years of age or older, (ii) met the Institutional Review Board (IRB) standard, and (iii) already had VP experience in Adult Nursing.

The Medical Foundation IRB in Taiwan (Number 202101569A3) approved the study, and organisational permission was obtained. The participants were informed that this study was voluntary, and they consented to be interviewed and audiotaped. The participants’ rights to privacy and anonymity were secured through the study’s procedure, and they could withdraw from the study at any time without penalty. Interview responses were confidential, and participants’ names could not be identified. All participants agreed to be identified by a prefix (e.g., SN for student nurse) and a number (e.g., SN1) in this publication.

### 2.3. Data Collection and Analysis

All individual interviews were conducted from September 2021 to January 2022. Purposive and theoretical sampling were utilised to recruit participants. The first two initial interviewees who met the recruiting criteria were purposively recruited. In addition, constant comparative data analysis was performed to focus interviews for the subsequent process of theoretical sampling, which aimed at developing the properties of categories until no new data emerged. The first author, an experienced qualitative researcher with PhD training and qualitative research publications, conducted the interviews in a private and quiet room at the university. The interview opened with the question, ‘Could you please share your experience with using virtual learning for your Adult Nursing practicum?’ Further questions were asked based on participants’ responses and an analysis of previous interviews. A semi-structured interview guide was used to avoid the interviewer’s personal interest in the topic and to maintain consistency. Individual face-to-face interviews were digitally recorded and transcribed into a text format immediately after the interview had finished. Data saturation was reached after 18 interviews. Each interview took about 40–60 min and resulted in thick data.

Initial, focused, and theoretical coding was performed [20]. The initial coding began by repeatedly reading the transcripts and coding each line. The line-by-line analysis enabled the researcher to interact with the data and sense what participants had said about the VP to obtain their perspectives on the VP’s overt and covert meaning. Each line of initial coding progressed to fulfil the fit and relevance of the study’s quality. Focused coding allowed us to identify those subcategories and their properties incisively and completely. The process of using focused coding to select codes specified possible relationships between categories to develop the theoretical coding and allowed us to move the study’s results in a theoretical direction.

The use of constant comparative data analysis provided an opportunity for the authors to interact and be involved with the data and to compare the data to find similarities and differences. The comparative method was used to construct distinctions at each level of the analytic work. The data were analysed separately by two researchers, and their analytic results were compared and discussed at regular meetings until a consensus was reached on the interpretation of the study’s insights. Memo writing was conducted to develop codes and redirect further data. It also provided a way of raising focused codes to conceptual categories to construct the core category of the study.

### 2.4. Trustworthiness

Attributes of credibility, originality, resonance, and usefulness guided the assessment of the study’s quality [20]. Credibility was ensured by having one researcher collect interview data. The purpose of this was to maintain the consistency and quality of the interview by using an interview guide. Further strategies employed to enhance credibility were to recruit as many participants as possible and to encourage participants to share their perspectives and experiences. Originality was reflected in the study result, which has provided theoretical significance regarding nursing students’ psychological resilience during the VP experience. All research members agreed on the study categories and read the report. Direct quotes used to present data in the findings section provided rich data for description and added validity to the originality of the study. Resonance was achieved in categories which provided meaningful data regarding students’ in-depth VP experiences, and the interviews were audiotaped and transcribed verbatim to ensure accuracy. The study results contributed to an improved knowledge of nursing students’ experiences during the VP, which can enhance preparations for future online learning. The study process was considered rigorous.

## 3. Results

The core category that emerged from the interview data was the psychological resilience associated with transitioning from an actual clinical practicum to VP during the pandemic through the process of the following three subcategories (Figure 1). All participants were full-time female undergraduate nursing students aged 20 to 21 years who shared a similar learning background.

### 3.1. Category 1: Learning Difficulties within One’s Inner Self

Most students participated in the VP despite doubts, curiosity, and a lack of VP experience to provide a frame of reference. During this process, three subcategories of students’ learning difficulties were identified with respect to their inner-world experiences during the VP.

#### 3.1.1. Learning Frustration

Students expressed frustration with the VP due to a lack of understanding of individual patients’ health status since it was difficult to make informed decisions and judgements about providing appropriate care. Given that there was a lack of medical equipment and supplies available for practicing their skills (during the VP), participants tended to remember and verbalise technical procedures. Hence, it was challenging to observe and respond appropriately to manage the patients. These difficulties lead to uncertainty, insecurity and learning frustration that in turn had an impact on the students’ learning and caused them to respond inappropriately.


*‘The patient’s health conditions were provided, but with inadequate information it was difficult to provide proper clinical judgement and decisions about care. I (as a student nurse) was unable to gain additional information from the virtual cases, since the information had been fabricated and imagined, except for the case health information provided by the clinical instructor. … This leads to hesitation in judging and recording patient care’. (SN1-1)*



*‘We did not always know what the next step was for our virtual practicum. When we asked for more information about this virtual case, we could not obtain the case’s health information fully... we worried we would make the wrong care decision for patients. We also worried about not meeting the teacher’s expectations and the learning standard with no previous experience to consult others about …no real patient to interact with and practice our care skills on…we imagined all the care situations, which led to learning uncertainty and insecurity’. (SN 1-2)*


This frustration had a negative impact on the students’ sense of security and confidence in their ability to learn.

#### 3.1.2. Learning Uncertainty and Insecurity

With VP-based learning, it was difficult for the students to understand what they learned and what they should know, retain, and remember for future practice. The VP relied on imagining and dictating steps for care technique practices and utilising creative materials to simulate medical equipment (mentioned in the subcategory of unleashing creative potential) to improve the learning experience. However, the students ended up feeling insecure and worried that their learning was inadequate because they were not practicing in a real care environment. SN2-1 mentioned:


*‘Me and the clinical instructor were talking at cross purposes. The instructor said to me, that was your patient, [and] … asked me to rethink the patient’s health status and proper care. However, despite my theoretical knowledge, I could not imagine a patient’s health status in advance. In my mind, I was at a loss to practice care techniques’.*


These feelings of learning uncertainty and insecurity lead to learning frustration, as SN 2-2 said: *‘The care techniques could not replicate real practice …I do not know what the patients in a certain situation were like…my memory had no unhealthy people…I do not know how to react to the patients and to connect this practicum with my future nursing learning’*.

#### 3.1.3. Venting Learning Nervousness

Students vented their tension and stress before being able to solve their learning difficulties. When they could not respond to questions from the clinical instructor or cope with their learning stress, they remained silent. Although saying nothing was an inappropriate response to learning frustration, it was a safe way for them to temporarily cope with their tension. SN3 said, *‘When the instructor asked questions and we did not know the answer, … everyone stayed silent as they did not have the courage to ask for more clues from the instructor. Crying was another way that the students released learning tension’*. SN4-1 said, *‘I was very sad, with tears [rolling] down my face, when I was scolded by the instructor for not working hard enough and for not having sufficient knowledge about nursing care. … I felt aggrieved*’.

The VP learning environment restricted the instructor and other staff, such as school counsellors, from guiding the students towards a more positive way of coping to release their learning tension/frustration. After venting their learning anxiety, students realised that it was caused by the VP learning environment, leading them to set more realistic goals for their practice in the VP. SN4-2 said, *‘The coronavirus outbreak forced our clinical placement to transition to the VP because of physical isolation for all parties’ health in the community...It also distanced me and other people. I had no one to talk about my learning difficulties and was unable to access others’*.

### 3.2. Category 2: Staying Positive and Confident

These two steps led students to stay confident and have a more positive outlook. ‘Fading the negative events’ shortened the time that students spent feeling down about their learning difficulties, whereas ‘determining self-learning’ from the virtual experiences improved their capacity to balance their psychological qualities and feelings associated with learning difficulties.

#### 3.2.1. Step 1: The Fading of Negative Events

**Convincing oneself.** The university opened up for students to decide whether to postpone their clinical placement or continue with the VP. An actual clinical placement would result in a postponed graduation and a future clinical placement with junior students. Most students chose the VP, but only later realised its limitations. As SN 5-1 said, *‘I have to take responsibility for deciding about the VP because I do not want to postpone my graduation…neither do I want to have an actual clinical placement with junior nursing students later if I do not take the VP. Other students could cope with the VP, and so can I’*.

Accordingly, participants within the VP had to adjust their expectations with respect to clinical placement. They convinced themselves to work harder to be better prepared for their later clinical placements. As SN5-2 said, *‘I would work harder and better prepare myself for the next clinical placement to do the actual care techniques with real patients. … Other students did finish their VP, and so could I’*.

It was found that the students chose not to focus on their learning difficulties and encouraged themselves to continue the VP. SN6 said, *‘The learning outcomes of the VP were different from the actual practicum. … still, I could accept that my learning was similar to that of the majority of the other students. I did not want to stick out or be behind the other students’*.

After the participants committed themselves to the VP, they learned how to work within the constraints of the VP.

**Ability to understand and accept.** Students felt that they were able to understand and accept their learning difficulties and restrictions. SN7-1 said, *‘The instructor utilised many web apps and web-learning platforms to help us learn, such as Slido, Google Jamboard, videos, etc. We were guided to learn in this way, as was the instructor, we thought’*.

Both the students and the instructor were unfamiliar with the new learning platform but understood that the VP was the only option for continuing their learning at that time. This, in turn, encouraged the students to pursue their own personal VP progress. Students understood and accepted the VP learning limitations, which lead to ‘a decisive awakening’ about ‘fading of negative events’. SN7-2 said, *‘Once we made a decision on the VP, we needed to move forward to see what we could learn from the VP....Immersing ourselves in the VP learning dilemmas does not help our practicums or change the status quo…We are already here. There is no reason to look back frequently and complain…’*.

**The progress of pursuing personal learning.** Students at this stage understood the circumstances and as they reluctantly accepted the new learning outcomes, they realised the restrictions of the VP. Accordingly, the students adjusted their expectations compared to the traditional clinical placements and set their new learning goals within the VP. They valued what they had learned from the VP and accepted that their practicum performance was different from the traditional clinical placement. Pursuing and progressing towards personal learning led students to emphasise their efforts to learn rather than focusing on learning difficulties. SN8-1 said, *‘The VP was the only option currently, but we could at least do something to improve our learning’*. SN 8-2 said, *‘When the VP is unchangeable, we need to do something for our VP learning… we need to make the period of VP time worthwhile …we set our own learning goal’*.

When participants were adjusted to the VP, they began to see what they had learned from the VP.

#### 3.2.2. Determining Self-Learning

As the students changed their mindsets, their expectations about learning within the VP became clearer and negative events faded. Consequently, they began to see what they had learned from the VP and what they could do to improve their learning. Students perceived that they improved their writing skills, professional knowledge, and creative ability.

**Improved writing skills.** Many students reported that their written nursing records and case reports had improved because of the online practicum, as more time was spent on writing and time was saved from actual patient care. When the instructor guided a student’s writing online, other students learned from this experience by example and quickly modified their own writing. Given that both students and instructors deepened their discussion of virtual patients’ conditions, this, in turn, strengthened the students’ understanding of the writing requirements and the important points to include in nursing records and case reports. SN9-1 said the following:


*‘When the instructor discussed someone’s work online, other classmates checked their own work to see whether or not they met the writing criteria and if not, instantly modified it. The instructor spent a lot of time correcting people’s work. We learned and improved our writing skills in this way’.*


SN 9-2 said, *‘During the clinical placement last time for fundamental nursing, my writing of nursing records was in a routine format which may not reflect the patient’s situation exactly. Now I know how to efficiently write the nursing record. I learned it from the clinical instructor when he/she discussed it with other learners online’*.

**Advance of theoretical knowledge**. The students expressed that they mainly learned theoretical knowledge, and they were concerned about this, since the focus of care activities and care for patients shifted from actual practice to verbalisation. SN10-1 stated, *‘Things were rushed, and it felt tense going from lecture to lecture, learning the theoretical knowledge of adult nursing. I didn’t have time to digest what I had learned by the time the next learning module started. During the VP, the theoretical knowledge and virtual cases were taught at the same time’*.

Yet, students reported that their knowledge of adult nursing had improved. SN 10-2 said: *‘The online practicum saved time from actual care and allowed more time to discuss theoretical knowledge related to similar cases. The clinical instructor asked plenty of questions … we did not worry about responding correctly, because the VP allowed us to search for answers online quickly in order to respond to the clinical instructor. We did not have questions after the VP time’*.

**Unleashing creative potential.** The VP inspired students’ creativity, as the online practicum meant that they were unable to provide nursing care involving real procedures with actual patients. As such, this interfered with getting the students’ learning needs met. Although using their imagination to care for patients was difficult for student nurses at first (due to their limited real-world clinical experience), care activities were identified and medical instruments were designed. SN11 remarked, *‘The aims of the clinical placement were to verify what care techniques and types of care were learned from the textbook’*.

Students therefore utilised their imagination to design medical objects and make supplies into models for their care practice. SN12 noted, *‘I made a model of a tracheal tube to practise stoma care of the tracheal tube’*. Another participant, SN13, commented, *‘I drew a wound on paper according to a description from an instructor, and I cut and placed it on the skin. I did the wet dressing for wound care in this way’*.

The medical supplies that students designed from their imagination were utilised in actual models of care to help them learn as they demonstrated their creative potential. SN14-1 concluded, *‘We made medical equipment we needed for a technique practice, such as an IV route and a bottle of fluid and a clothes hanger as an IV stand... we made any needs for our learning. We created a learning environment as the clinical ward for our VP learning. We shared these creative features with others.’*

### 3.3. Category 3: Knowing What Is Possible

Through subcategory 2, ‘staying positive and confident’, students came to know what things were replaceable and irreplaceable in the virtual practicum. Students valued their unique experiences and did not feel learning frustration, as they pursued a personal psychological balance in VP learning.

#### 3.3.1. Learning (In)Effectiveness

The VP replaced the traditional clinical placements that students normally undertake. Without the actual practice, experience and interactions with patients, the students reported that the training was not as comprehensive or adequate for learning clinical judgement, observational skills, and empathy for the patients. Said SN14-2, *‘We accepted the changes compared to the clinical placement when replaced by the VP and recognised both the positive learning effects and limitations of undertaking the VP. Despite technological advances, it cannot replace interactions with real people. Certain types of learning need to be acquired through interactions with individuals, such as empathy, interpersonal relationship skills and non-verbal behaviour. Also, the care techniques could not be practiced appropriately’*.

On one hand, the benefits of the VP were demonstrated in category 2, ‘staying positive and confident’, yet on the other hand, the learning limitations of the VP worried students.

#### 3.3.2. Worry

Students reported being worried about their skill levels and that they would be incompetent in future work settings in adult nursing. As such, the VP was perceived with a negative first impression compared to a traditional clinical placement. SN15-1 said, *‘I lacked a reference for future clinical work, as I missed the actual practicum of Adult Nursing. For example, I did not have actual work experience in adult nursing. I may be incompetent for a role in adult nursing, so I might not choose adult nursing for my future career’*.

Students worried about the future work domain associated with the VP. This latent concern reflected their dissatisfaction with the VP learning method. SN 15-2 said, *‘I had tried my best to learn from the VP, but I did not have confidence for my future practicum and worried about my future career because learning adult nursing is the basis of all care domains*’.

#### 3.3.3. Valuable Experiences

Students expressed that they had valuable experiences during the VP. This included what they had learned and the effectiveness of the learning. Students also expressed that the practicum in clinical settings provides hands-on experiences that cannot be replaced by the VP, as a clinical practicum helps consolidate theoretical knowledge with clinical skills. This was expressed as the bodily memory of learning, which is a learning process of internalisation. SN16 noted, *‘My body would memorise what I had been through in this experience. … The learning and memory of the VP were superficial and temporary; therefore, the time for the learning retention rate was short. I’m worried the VP’s learning outcomes won’t be retained for a long time’*.

As such, ongoing VPs could potentially cause learning insecurity for students in the long term. SN 17: *‘Although I had learned something from the VP and this experience was unique, I would not go through an online practicum again. If the coronavirus disease continues to affect our daily lives and learning, I would like to postpone my next clinical placement for a real clinical practicum. I still appreciated the unique experience of the VP’*.

## 4. Discussion

The present study identified psychological resilience as the process of coping with and modifying thoughts and feelings associated with the students’ learning difficulties while undertaking the VP. This capacity for psychological resilience led students to become more aware of their challenges and being better able to cope with them and manage to stay more optimistic and ‘draw victory from adversity’. As such, psychological resilience provided students with opportunities for continuing professional development and learning rather than being overwhelmed by anxiety, insecurity, and stress, which is similar to Kunaviktikul et al.’s [22] findings. To have a positive mindset and thinking originating from one’s inner self is a key constituent to strengthen psychological resilience [23] and to interact with the social world confidently [24]. To stay positive and confident in the current study was recognized as fading negative events and realizing the advantageous occurrences [23]. Fading negative events reduces undesirable emotions related to difficulties and leads to optimism, learning efficacy and not ignoring but dealing with potential problems [25].

Psychological resilience emphasizes individuals’ ability to cope with adversity and to continue to grow and evolve from adversity, while resilience means the ability for an individual’s mind and feelings to bounce back when facing difficulty [26]. After a reflection on and adjustment to the learning difficulties of the VP, students experienced resilience [24], reduced learning stress [27] and improved motivation for self-directed learning [28]. Hence, participants tended to be more self-directed, accepted learning restrictions and adapted their expectations of the VP experience, which is in line with Hunter Revell et al. [24] findings suggesting that students’ resilience helped their learning experience even though the VP was difficult. Thus, there is an important need for undergraduate nursing students to develop resilience and skills to cope to reduce learning difficulty [29] which corresponds to the view of Paoletti et al. [30] to reorganize a ‘new normality’ for the post pandemic of everyday life.

Overall, the coronavirus pandemic as an ongoing traumatic event significantly affected students’ learning [31] and more specifically impacted the clinical Adult nursing placement that led to the implementation of the VP. This in turn interrupted traditional learning strategies and the environment of didactic lectures increasing students’ learning frustration. As the VP became the only viable option of continuing clinical practice, this added to students’ learning pressure, in the form of learning insecurity. This finding in the current study is in accordance with Collado-Boira et al.’s results [16]. Learning insecurity can be overcome for students by increasing their familiarity with and their understanding of the utilized teaching–learning strategies [16] increasing their resilience during the VP. Furthermore, psychological resilience may also assist students in continuing their professional development associated with an improved capacity for self-regulation, knowledge, and coping skills to deal with unfamiliar learning environments. As such, it is important to teach students such coping skills to better manage stress and learning anxiety.

Within the category of ‘staying positive and confident’, this study found that students managed to adjust to their rapidly changing learning environment by ‘fading the negative events’ and ‘determining what oneself learned’. The subcategory of ‘fading the negative events’ began with convincing oneself to undertake the process of changing and interacting with the online VP world as part of their learning environment. This online learning world included both new opportunities such as learning in a vast online environment as well as limitations including students’ interactions with the clinical instructor and peers via a computer screen, leading to a sense of isolation. Being in this isolated learning world restricted individuals from dealing with others in person and the opportunities that come with learning from others in the real world, yet it improved individual’s capacity to think about, and how to pursue personal learning progress. Previous research by Lovri’c et al. [32], Wallace et al. [33], Suliman et al. [34] and Kunaviktikul et al. [22] contradicted our current findings that online learning had a negative impact on learning and psychological well-being and found that online learning was a positive experience for students. However, the process of ‘staying positive and confident’ did not mean that the students in our study were completely satisfied with their VP experience or experience of psychological well-being; rather, it was a process of developing psychological resilience within a difficult learning environment to continue their professional development and training.

The subcategory of ‘determining what oneself learned’ described what individuals learned in a positive way and constituted students’ psychological resilience. Students perceived each learning goal as built upon a series of lessons, and although those learning achievements was somewhat limited compared to didactic lectures, they still presented opportunities for increased learning. This process of pursuing learning effectiveness highlighted the students’ learning of resilience from the VP and enhanced their psychological resilience. Taken together, our study findings emphasized that the process of psychological resilience was constructed, which differentiated from a study targeting the resilience factors of online learning [22] or satisfaction as an effective alternative to online learning [32].

The category of ‘knowing what is possible’ led students to perceive learning in the social (clinical) world clearly and to understand what is possible/impossible or replaceable/irreplaceable after interacting with the social world. Although the VP could not replace the actual clinical placement completely, students understood the strengths and weaknesses of the VP learning. Students became conscious of planning for their future clinical placement, which was seen as self-directed learning. The self-directed component of future clinical placements is similar to Hunter Revell et al.’s [24] findings, in which students self-directed their learning, accepted restrictions and modified their personal achievements during the clinical experience. The experience of psychological resilience led students to be more confident with their clinical experience during the VP. Furthermore, Bozkurt et al. [35] advocated that the coronavirus pandemic may be an opportunity to transform teaching and learning mode to more creative pedagogy.

## 5. Study Limitations

This study constructed a substantive theory to understand student nurses’ psychological resilience undertaking a VP during the coronavirus pandemic. Although data saturation was achieved and the recruited participants indicated that their psychological well-being increased during the VP, students who were not involved (in this particular VP) may have had a different experience compared to our current students. Given that the research group mainly consisted of females, further research is needed to explore potential differences and similarities among males utilizing a larger research group than our current study. Another limitation was the recruitment of research group from only one university that may have resulted in a positive outcome. Thus, future research may benefit from a larger multisite qualitative study in concert with a survey-based approach to complement our understanding of students’ experiences and psychological well-being in relation to resilience in future educational pursuits within a VP context.

## 6. Conclusions

This study has provided evidence on the construction of psychological resilience among nursing students’ undertaking a VP during the coronavirus pandemic. During this process students recognized their learning potential and discovered their psychological resilience while adjusting to the rapid changes in this new learning environment. The study’s findings highlighted various learning difficulties tied to the clinical placement in the VP. However, the substantive theory of psychological resilience, staying positive and confident and knowing what is possible provided evidence that the students were able to cope with the adversity of the VP learning. As such, the students experienced the VP process; they became resilient and developed coping skills linked to their psychological resilience that they can utilize in future clinical placements and practical settings. These skills will also help maintain the students’ mental health and future learning in the nursing profession.

## 7. Relevance for Clinical Practice

Taken together, the results of this study provided a deeper understanding of the learning difficulties that student nurses encountered during the VP and highlighted the need to prepare for the ongoing pandemic as well as for future circumstances if another pandemic were to occur in the future. The substantive theory of psychological resilience highlighted a need for further training workshops on topics such as mindfulness, coping skills, reflective ability and/or personal development to assist student nurses when placed under similar adverse circumstances. It also provided a framework for improving student nurses’ mental health and maintain psychological well-being when dealing with challenging external factors while undertaking their VP as well as future real world clinical placements. Importantly, nursing education fosters competences, and further training should focus on recognizing and developing an individual’s learning potential. Thus, learning psychological resilience ensures a more enjoyable and smooth learning experience.

## Figures and Tables

**Figure 1 ijerph-20-01264-f001:**
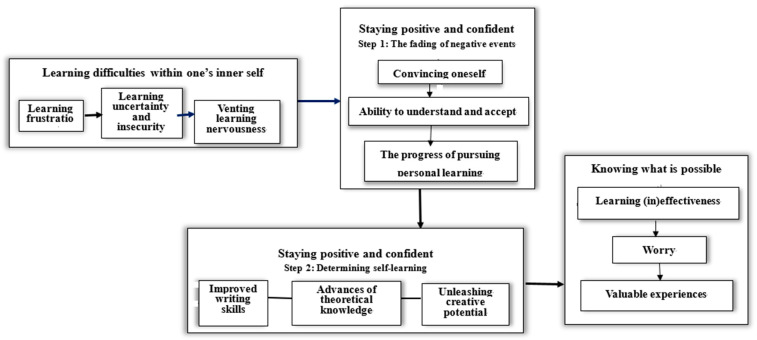
The process of developing psychological resilience in a virtual practicum.

## Data Availability

The data that support the findings of this study are available on request from the corresponding author, C.-C.L.

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
