# Peer review of "An Exploration of Psychological Resilience among Undergraduate Nursing Students Undertaking an Adult Nursing Virtual Practicum during the Coronavirus Pandemic in Taiwan: A Qualitative Study"

_ijerph, 2023, doi:10.3390/ijerph20021264_

Round 1

Reviewer 1 Report (Previous Reviewer 2)

The paper can be accepted as it is. They answered my comments

Author Response

Thanks reviewer 1. Please consult to the attached file.  

Reviewer 2 Report (New Reviewer)

·        “The COVID-19 (coronavirus disease of 2019)” I assume that “19” doesn’t originate from 2019, it would be better to revise as “The coronovirus pandemic (COVID- 19)…

·        I suggest authors to cite and provide a synthesis of the related literature, this section can be expended to create a sold intellectual background.

·        “Furthermore, all levels of Taiwan’s education system transitioned from teaching and 48 learning in person to online learning…” Authors can refer to “emergency remote Education” to better contextualize what has happened during the early waves of the pandemic.

·        “Learning environment changed made students fearful when…” Please pğay attention that there is a different use of font, better to stick to MPDI style.

·        In qualitative studies, we don’t prefer using the term “sample, because we don’t aim to generalize our findings. Instead, use “research group”. E.g. “Sample recruiting criteria were:..” Please address all instances.

·        Perhaps, authors can improve the suggestions and implications of the study especially in terms of educational perspectives.

·        Please see the following references, I belive that they would be helpful support your arguments.

·        Critical change in the educational landscape: Reimagining, reengineering, and redesigning a better future. Open/Technology in Education, Society, and Scholarship Association Conference Proceedings (16-20 May, 2022, Canada), 2(1), 1-8. https://doi.org/10.18357/otessac.2022.2.1.218

·        Resilience in a complex and unpredictable world. Journal of Contingencies and Crisis Management, 25(3), 118-122. https://doi.org/10.1111/1468-5973.12177

·        Reimagining the new pedagogical possibilities for universities post-Covid-19. Educational Philosophy and Theory, 1-44. https://doi.org/10.1080/00131857.2020.1777655

·        What can we learn from the Covid-19 pandemic? Resilience for the future and neuropsychopedagogical insights. Frontiers in Psychology. https://doi.org/10.3389/fpsyg.2022.993991

·        In all, I liked reviewing the manuscript and the above suggestions can improve its overall quality. I would like to see the manuscript being published after these minor revisions.

Author Response

Thanks to the reivewer 2 and please consult to the attached file for responses. 

This manuscript is a resubmission of an earlier submission. The following is a list of the peer review reports and author responses from that submission.

Round 1

Reviewer 1 Report

I thank the authors for the opportunity to review this interesting article, I think it is a relevant topic, but it presents important inconsistencies between the study design and the data collection tools, which influence in the results

In the study they use individual interviews in data collection and state that they used Constructivist grounded theory for understanding of social phenomena. Please explain to me how they were able if there was no interaction between the students and there was no social construction of the phenomenon.

He states that the interviews started with a question and the rest of the questions were clarifying the first. How were the interviews conducted? Faced with such a general question, the answers would be very different. In fact, this can significantly condition data saturation. Please detail this.

It would be interesting if they added a table of sociodemographic data in the results section. At the same time, it would be interesting if they accompanied the writing of the results with more narratives (verbatims) from the students, in order to give solidity to their affirmations.

All the best

Author Response

Reply as attached file

Author Response

Reply as attached file-response to reviewer 2. 

Round 2

Reviewer 1 Report

I consider that the argument to justify the methodology is not supported by the data collection tool. The interaction between professors and students does not justify the coherence between the methodology and the results, it continues to be an important weakness and I consider that it should not be published.